computational chemistry/analytical chemistry

oxybutynin hydrochloride, eosin Y, DOE, green

**Author for correspondence:**
Heba Elmansi
e-mail: dr_heba85@hotmail.com

This article has been edited by the Royal Society of Chemistry, including the commissioning, peer review process and editorial aspects up to the point of acceptance.

# Factorial design-assisted spectroscopic determination of oxybutynin hydrochloride

Eman Yosrey, Heba Elmansi, Zeinab A. Sheribah and Mohamed El-Sayed Metwally

Department of Pharmaceutical Analytical Chemistry, Faculty of Pharmacy, Mansoura University, Mansoura 35516, Egypt

HE, 0000-0002-3953-7169

In this study, we have developed two facile spectroscopic methods for quantifying oxybutynin (OBT) hydrochloride in its pure form and tablets using design of experiments (DOEs). The spectroscopic methods depended on the ion-pair complex formation between the tertiary amino group in the drug and eosin in 0.2 M acetate buffer of pH 4. Method I involves spectrophotometric measurement of the absorbance of the developed complex at 550 nm and showed linearity through 1.0–10.0 µg ml$^{-1}$. Method II involves spectrofluorometric measurement of the quenching influence of OBT on the native fluorescence of eosin (λ excitation/λ emission of 304/548 nm) and showed linearity through 1.0–6.0 µg ml$^{-1}$. Critical parameters were identified through preliminary trials and optimized using the DOE. Additionally, the quenching mechanism was investigated and the pathway of the reaction was postulated. The fluorescence quenching constant and thermodynamic parameters were explored using the Stern–Volmer plot and Van't Hoff graph, respectively. Assessments conducted via analytical ecoscale revealed the 'excellent-greenness' of the methodology. The two methods have the potentials of being green and fast compared with other reported methods.

## 1. Introduction

Overactive bladder (OAB) is a chronic medical condition in which patients suffer from 'urinary urgency, with nocturnal enuresis, with or without urgency incontinence, where there is no urinary tract infection or other obvious pathology' [1].

Oxybutynin (OBT) hydrochloride (figure 1*a*) 4-(diethylamino) but-2-ynyl (RS)-2-cyclohexyl-2-hydroxy-2-phenylacetate hydrochloride [2] as an anticholinergic medication is recommended for patients with OAB or detrusor overactivity symptoms, including urinary frequency and urgency [3]. Various analytical assays were

**Figure 1.** Structural formulae of: (*a*) OBT and (*b*) EY.

published to quantify OBT in its pure and dosage form. The BP pharmacopeia suggested a high-performance liquid chromatography (HPLC) method using C8 column and mobile phase of phosphate buffer, containing potassium dihydrogen phosphate and dipotassium hydrogen phosphate, added to acetonitrile (49 : 51) [2]. Other reported methods include spectrofluorometry, spectrophotometry [4–6], voltammetry [7] and chromatography [8,9]. These methods are either time-consuming, tedious or use expensive organic solvents harmful to the environment. Since green analytical chemistry has gained considerable attention in the field of pharmaceutical analysis, the current study aimed to provide simple and sensitive methods through using green reagents [10]. The methods relied on the formation of an ion-pair complex between eosin Y (EY) and OBT. Additionally, the experimental conditions were optimized using the design of experiments (DOEs). Some amino compounds can be determined using an acidic dye called EY as an ion-pairing agent (figure 1*b*) via binary complex formation [11–15]. DOE is a powerful data collection and analysis tool that can be used under various experimental conditions [16]. The interest in DOE originates from its ability to conclude the most effective factors, put them in order and study the interaction between these factors, which cannot be studied in one factor at a time. This information is given through optimization plots and other charts, which improve the experimental parameters and method validation [16]. Quenching mechanism of the spectrofluorometric method was investigated using Stern–Volmer equation. Changes in enthalpy $\Delta H°$, entropy $\Delta S°$ and Gibbs free energy $\Delta G°$ are significant thermodynamic parameters in determining the binding forces in the produced complex [17–23]. The greenness of the proposed methods was assessed by analytical ecoscale and showed higher score than the BP reference method [2].

Up till now the use of DOE was not yet discussed for OBT determination. In this study, DOEs was used as a statistical method for assessing the influence of critical parameters on the proposed methodology to save time, effort and chemicals. The suggested procedure presented a sensitive way for quantifying OBT using inexpensive reagent, simple and economic instruments.

# 2. Experimental

## 2.1. Apparatus

— Shimadzu ultraviolet–visible (UV–Vis) 1601 recording spectrophotometer (P/N 206–67001) was used. Recording range: 0–1.0 at a wavelength of 550 nm.
— Cary Eclipse fluorescence spectrophotometer with a xenon flash lamp, from Agilent technologies, was used at medium voltage mode (530 V). The spectra were recorded at 548 nm after excitation at 304 nm using a smoothing factor of 20.
— For the reference BP method, Shimadzu Prominence HPLC system (Shimadzu Corp., Kyoto, Japan) with an LC-20 AD pump, DGU-20 A5 degasser and SPD-20A UV–Vis detector.

## 2.2. Materials and reagents

— OBT hydrochloride was acquired from the Egyptian Company for Chemical and Pharmaceuticals (ADWIA) (10th of Ramadan City, Egypt) with a purity of 100.06% as certified.
— Uripan tablets, batch no. 2009101, containing 5 mg OBT per tablet, a product of the ADWIA Co.S.A.E, 10th of Ramadan City, Egypt was purchased from the local pharmacy.
— Eosin Y (Riedel-de-Haën Seelze, Germany) was prepared as $4 \times 10^{-3}$ M and $5 \times 10^{-4}$ M aqueous solutions for the spectrophotometric and the spectrofluorometric methods, respectively.
— An acetate buffer solution (pH range from 3.5 to 5.5) was prepared by mixing 0.2 M sodium acetate and 0.2 M acetic acid (El-Nasr Pharmaceutical Chemicals Company, Egypt).
— Surfactants (El Nasr Chemical Co. Egypt) were prepared at 1.0% w/v in distilled water, such as cetrimide, Tween 80 and sodium dodecyl sulfate (SDS).

## 2.3. Standard solution

Distilled water was used during the study.

An aqueous stock solution of 100 µg ml$^{-1}$ OBT was prepared by transferring 10 mg OBT to a 100 ml volumetric flask, approximately 50 ml distilled water was added and sonicated for 15 min and completed with distilled water to the final mark. Dilutions were made to obtain the required concentrations using the same solvent. The solution was stored in the refrigerator and protected from light to attain maximum stability.

## 2.4. General procedure

### 2.4.1. Procedures for calibration curves

#### 2.4.1.1. Method I: spectrophotometric method

A series of different volumes from OBT stock solution were added to 10 ml volumetric flasks to obtain a concentration range from 1.0 to 10.0 µg ml$^{-1}$, approximately 4.0 ml of distilled water were added followed by 0.5 ml of EY ($4.0 \times 10^{-3}$ M); 0.5 ml of 0.2 M acetate buffer (pH 4) was added and mixed well. The flask was completed with distilled water to the mark. The precipitation of the formed complex, particularly at a high concentration of EY, was overcome by following this order of addition [15]. Absorbance (ABS) was recorded at 550 nm in parallel with the blank experiment. The ABS of the formed complex was plotted against the final concentration (µg ml$^{-1}$) and the regression equation was concluded.

#### 2.4.1.2. Method II: spectrofluorometric method

The sequence of previous steps was followed within the concentration range (1.0–6.0 µg ml$^{-1}$) of OBT but with 1.0 ml of EY ($5.0 \times 10^{-4}$ M). After excitation at 304 nm, the fluorescence intensity of the complex ($F$) was recorded at 548 nm. The quenching values ($\Delta F$ = fluorescence intensity EY solution ($F^{\circ}$) – $F$) of the prepared solutions were measured. To construct the standard calibration graph, $\Delta F$ was plotted against the concentration (µg ml$^{-1}$) and the corresponding regression equation was concluded.

### 2.4.2. Procedure for OBT assay in dosage form

Ten tablets of Uripan were weighed, grounded and mixed. An accurately weighed quantity of the powder, containing 10.0 mg OBT, was dissolved in approximately 50 ml of distilled water in a 100 ml volumetric flask. The flask was sonicated for 30 min and filled up the mark with distilled water. The solution was filtered using double-filter papers Whatman (grade 1), rejecting the first part of the filtrate. Aliquots from the filtered solution were transferred to 10 ml volumetric flasks, adopting either spectrophotometric or spectrofluorometric procedures. The content of OBT in its tablet was calculated using a regression equation.

### 2.4.3. Procedure for determining the reaction stoichiometry via Job's method

Equimolar solutions of EY and OBT were prepared as $4 \times 10^{-3}$ M. In 10 ml volumetric flasks, different molar ratios of the drug and reagent were taken, keeping a fixed total molar concentration. The values of the obtained ABS were plotted against mole fractions of OBT ([OBT]/[OBT] + [EY]).

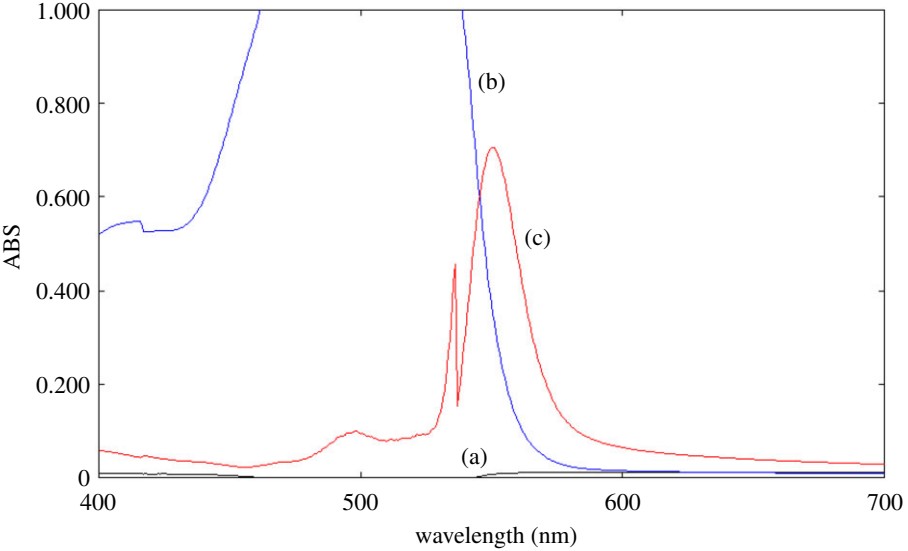

**Figure 2.** Absorption spectra of: (a) OBT in aqueous solution (8.0 µg ml$^{-1}$) in acetate buffer of pH 4; (b) blank EY ($4 \times 10^{-3}$ M) in acetate buffer of pH 4; (c) the reaction product of OBT (8.0 µg ml$^{-1}$) with EY ($4 \times 10^{-3}$ M) in acetate buffer of pH 4.

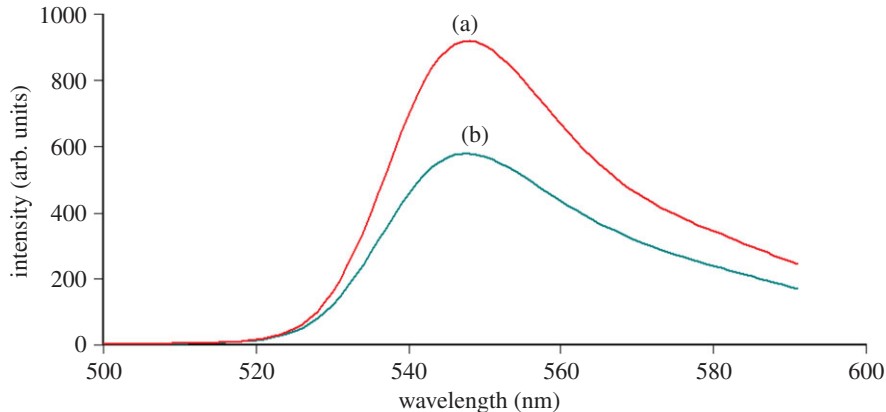

**Figure 3.** Emission spectra of: (a) blank EY ($5.0 \times 10^{-4}$ M) in acetate buffer of pH 4; (b) reaction product of EY ($5.0 \times 10^{-4}$ M) and OBT (4.0 µg ml$^{-1}$) in acetate buffer of pH 4.

# 3. Results and discussion

The proposed methods aimed at being green, sensitive, rapid and economical. The developed spectroscopic methods using EY attained these characteristics in addition to selectivity.

EY is a carboxylic dye that is used as an ion-pairing reagent for the spectroscopic determination of cationic drugs in an acidic medium by forming ion-pair complexes [11–15]. In the spectrophotometric method, the ABS of the developed complex was measured at 550 nm (figure 2). In the absorption spectrum, blank EY in acetate buffer at pH 4, the maximum ABS is more than 1. This is explained by: theoretically, since ABS = −log transmittance, ABS value is ranged from 0 to 2; but practically, ABS value is selected within 0.2–1.2 to avoid instrumental errors [24]. In the spectrofluorometric method, the binary complex was measured as fluorescence quenching of EY at 548 nm, after excitation at 304 nm (figure 3).

## 3.1. Optimizing the experimental conditions

All parameters that affect the development and stability of the produced complex between EY and OBT were studied and optimized using DOE. These parameters include buffer pH and volume, EY volume, reaction time, temperature effect and diluting solvent.

### 3.1.1. pH and volume of buffer

The pH of the medium is studied because the ionized EY is required for a complete reaction with the drug [25]; 0.2 M acetate buffer was used to study the different pH range (3.5–5.5) to attain maximum ABS. The ABS at pH between 3.5 and 4 was the highest (electronic supplementary material, figure S1). Different volumes (0.2–2.0 ml) of acetate buffer (pH = 4) were then studied; volume between 0.5 and 1.0 ml was the optimum.

### 3.1.2. Volume of EY

Different volumes of EY, $4 \times 10^{-3}$ and $5 \times 10^{-4}$ M were studied for spectrophotometric and spectrofluorometric methods, respectively. We found that 0.5–1.0 ml of ($4 \times 10^{-3}$ M) EY was required to attain maximum ABS (electronic supplementary material, figure S2) and 1.0 ml of ($5 \times 10^{-4}$ M) EY to achieve maximum $\Delta F$ (electronic supplementary material, figure S3).

### 3.1.3. Effect of time

The effect of time on the reaction between EY and OBT was examined. The formation of the complex was immediate, and the ABS was almost constant for 30 min.

### 3.1.4. Diluting solvents

Several diluting solvents were studied to obtain the highest sensitivity of the developed methods: distilled water, methanol, ethanol and acetonitrile. Distilled water was chosen because it gave maximum ABS. The ABS was decreased using organic solvents. Using distilled water offered greenness and economical advantages to the proposed methods.

### 3.1.5. Effect of surfactant

Effect of surfactant was investigated using 1.0% w/v of SDS, cetrimide and Tween 80. No enhancement was attained, therefore, no surfactant was used.

### 3.1.6. Effect of time and heating

The effect of different periods 5.0–30.0 min at 50°C was studied. The ABS decreased up to 30 min and no enhancement was noted, so the two methods were performed at room temperature adding time-saving and greenness proprieties.

### 3.1.7. Design of experiments

From the previous studies, it was deduced that three independent factors influence the dependent response. These factors include the pH of the reaction (pH, A), volume of acetate buffer (V.ACE, B) and volume of EY reagent (V.R, C). The dependent response was the ABS of the formed complex; $2^3$ FFD (full factorial design of three independent factors at two levels) was chosen to optimize reaction conditions. The pH, 3.5 and 4, were chosen as the two levels of the pH factor because the ABS was nearly maximum at that range. The two levels of V.ACE at which the ABS was constant and maximum were 0.5 and 1.0 ml. The two levels for the V.R were 0.5 and 1.0 ml for the same reason.

Eight designs were done according to $2^3$ FFD to exclude optimum conditions that gave maximum response. The most important advantages of using DOE include identifying the most influencing factors affecting the reaction and putting them in order. Additionally, studying the interactions between these factors that cannot be studied using the common optimization work, as one factor is changed while keeping the other factors constant [16]. Minitab software uses the input variables to get the target. In addition, it uses values extracted to indicate to what extent those settings fulfil the response target. Optimization is measured by the composite desirability ($D$), which has a range of 0–1. High value of composite desirability indicates the conditions achieve the target. The software extracts the optimal conditions through the optimization plot (figure 4) with the desirability value. The extracted optimum conditions include pH of 4, 0.5 ml buffer volume and 0.5 ml EY reagent to attain maximum ABS (table 1 and electronic supplementary material, table S1). FFD provides mathematical evaluation for ABS response using the estimated coefficients of the independent factors (data in coded units) (electronic supplementary material, table S2).

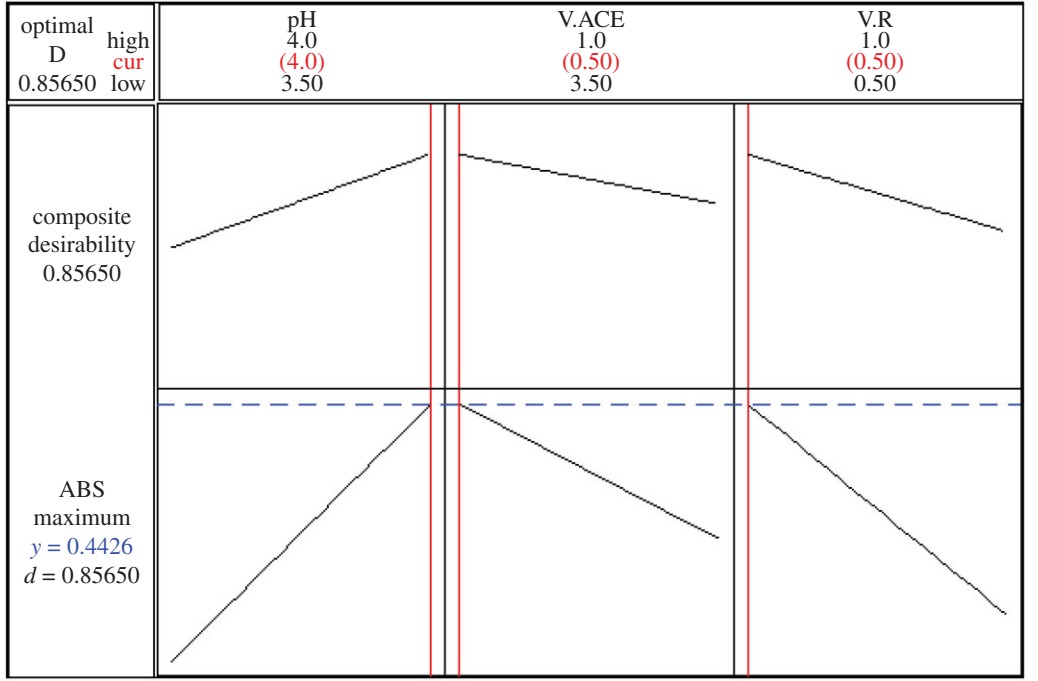

**Figure 4.** $2^3$ full factorial design optimization plot for the reaction between OBT and EY (where ABS: absorbance of the produced complex, pH: pH of the medium, V.ACE: volume of acetate buffer and V.R: volume of $(4 \times 10^{-3}$ M) EY).

**Table 1.** Response optimization of $2^3$ factorial design for absorption of EY−OBT complex.

| parameters | global solution: pH = 4, V.ACE = 0.5, V.R = 0.5 | | | | | | | |
|---|---|---|---|---|---|---|---|---|
| | goal | lower | target | upper | weight | import | predicted responses | desirability |
| ABS[a] | maximum | 0.1 | 0.5 | 0.5 | 1 | 1 | 0.4426 | 0.85650 |

[a]Where ABS: absorbance of the produced complex, pH: pH of the medium, V.ACE: volume of acetate buffer and V.R: volume of $(4 \times 10^{-3}$ M) EY.

According to the Pareto plot of factor effects on the ABS (electronic supplementary material, figure S4), (C) factor has the strongest impact on the ABS, however, it is not significant for a 95% confidence level. The strongest impact was confirmed by the highest slope in the main effects plot (electronic supplementary material, figure S5). From the normal plot of the effects (electronic supplementary material, figure S6) and estimated effects (electronic supplementary material, table S2), this factor impacts negatively on the reaction, meaning that as the volume of the reagent increases ABS decreases.

From the interactions plot (electronic supplementary material, figure S7), (C) factor affects negatively on ABS when interacting with (A) or (B), either at upper or lower levels for both factors (A and B). (B) factor impact negatively on ABS when it interacts with factor (A) or (C) at upper levels, for both factors, however, at the lower levels, for both factors, there was a little positive influence. (B) factor has the smallest negative impact on ABS, as shown in the Pareto plot (electronic supplementary material, figure S4). It was confirmed by the smallest slope in its corresponding curve in the main effect (electronic supplementary material, figure S5) and the lowest value in estimated effects (electronic supplementary material, table S2). From the interactions plot (electronic supplementary material, figure S7), (A) factor has a positive influence on ABS when it interacts with (B) or (C) factors, either at the upper or lower levels, for both factors. From AB and AC interactions, it was recognized that the strong positive influence of the (A) factor on ABS diminished at either the lower level of the (C) factor or upper the level of the (B) factor.

We deduced from the Pareto plot (electronic supplementary material, figure S4) that the (AC) interaction is the most important interaction factor. The interaction plot (electronic supplementary material, figure S7) emphasizes this interaction through the unparalleled lines.

**Table 2.** Performance data of OBT hydrochloride by the proposed methods.

| parameter | spectrophotometric method | spectrofluorometric method |
| --- | --- | --- |
| concentration range (µg ml$^{-1}$) | 1.0–10.0 | 1.0–6.0 |
| LOD (µg ml$^{-1}$) | 0.14 | 0.08 |
| LOQ (µg ml$^{-1}$) | 0.41 | 0.25 |
| correlation coefficient (r) | 0.9999 | 0.9999 |
| slope | 0.08 | 61.15 |
| intercept | 0.06 | −37.66 |
| $S_{y/x}$ | 0.0038 | 1.38 |
| $S_a$ | 0.0031 | 1.50 |
| $S_b$ | 0.0005 | 0.36 |
| % error | 0.55 | 0.43 |
| % RSD | 1.22 | 0.96 |
| no. of experiments | 5 | 5 |
| mean found (%) ± s.d. | 100.09 ± 1.22 | 100.18 ± 0.96 |

$S_{y/x}$ = standard deviation of the residuals. $S_a$ = standard deviation of the intercept of regression line. $S_b$ = standard deviation of the slope of regression line. % Error = RSD%/$\sqrt{n}$.

## 3.2. Method validation

The developed methods were validated following the International Conference on Harmonization (ICH) recommendations [26]. The defined calibration ranges were used to check the linearity, specificity, accuracy and precision procedures of the proposed methods (table 2).

### 3.2.1. Linearity and concentration range

In the spectrophotometric method, the OBT calibration curve was obtained by plotting the ABS values versus its corresponding concentration. Linearity was achieved within the range of 1.0–10.0 µg ml$^{-1}$. In the spectrofluorometric method, the OBT calibration curve was obtained by plotting fluorescence quenching ($\Delta F$) value versus its corresponding concentration, showing linearity within the range of 1.0–6.0 µg ml$^{-1}$. Linear regression parameters are illustrated in table 2. The negative intercept obtained by the spectrofluorometric method is expected to be due to interference from matrix components which may be due to OBT itself.

### 3.2.2. Repeatability and intermediate precision

Three concentration levels of OBT were investigated three times during the same day or through three successive days using general analytical procedures (table 3). The methods result in a satisfactory % RSD and % error, indicating acceptance repeatability and intermediate precision for the proposed methods. Intermediate precision (also known as ruggedness) expresses within-laboratory variation, as on different days, or with different analysts or equipment within the same laboratory. Repeatability refers to the use of the analytical procedure within a laboratory over a short period of time using the same analyst with the same equipment

### 3.2.3. Limit of quantitation and limit of detection

Limit of quantitation (LOQ) and limit of detection (LOD) were determined according to ICH recommendations [26] as follows:

LOQ = 10 × intercept standard deviation/slope

LOD = 3.3 × intercept standard deviation/slope

LOD has the value of 0.14 and 0.08 µg ml$^{-1}$ for Methods I and II, respectively, while LOQ has 0.41 and 0.25 µg ml$^{-1}$ for Methods I and II, respectively.

### 3.2.4. Robustness

It was confirmed by getting steady readings through small variability in method parameters. The evaluated variables, such as pH, the volume of acetate buffer and EY volume, revealed that the proposed methods were robust through routine work.

**Table 3.** Precision data of the proposed methods for determination of OBT hydrochloride in pure form.

| spectrophotometric method | | | spectrofluorometric method | | |
|---|---|---|---|---|---|
| sample concentration | repeatability | intermediate precision | sample concentration | repeatability | intermediate precision |
| 2.0 µg ml$^{-1}$ | | | 3.0 µg ml$^{-1}$ | | |
| mean found (%) $\bar{x}$; | 101.75 | 101.75 | $\bar{x}$; | 99.30 | 100.39 |
| ± s.d. | 0.38 | 0.38 | ±s.d. | 0.33 | 0.62 |
| % RSD | 0.37 | 0.37 | % RSD | 0.33 | 0.62 |
| % error | 0.22 | 0.22 | % error | 0.19 | 0.36 |
| 6.0 µg ml$^{-1}$ | | | 4.0 µg ml$^{-1}$ | | |
| mean found (%) $\bar{x}$; | 100.35 | 101.15 | $\bar{x}$; | 99.24 | 100.11 |
| ±s.d. | 1.33 | 1.26 | ±s.d. | 0.47 | 1.13 |
| % RSD | 1.33 | 1.25 | % RSD | 0.48 | 1.13 |
| % error | 0.77 | 0.72 | % error | 0.28 | 0.65 |
| 8.0 µg ml$^{-1}$ | | | 6.0 µg ml$^{-1}$ | | |
| mean found (%) $\bar{x}$; | 99.96 | 99.96 | $\bar{x}$; | 98.58 | 99.77 |
| ±s.d. | 0.95 | 0.95 | ±s.d. | 0.30 | 1.33 |
| % RSD | 0.95 | 0.95 | % RSD | 0.31 | 1.33 |
| % error | 0.55 | 0.55 | % error | 0.18 | 0.77 |

### 3.2.5. Accuracy

The calculated values from Student's $t$-test and variance ratio $F$-test were lower than the tabulated ones, indicating no significant difference between the two proposed methods and reference method [2]. In addition, it shows the accuracy of the two methods [27] (table 4).

## 3.3. Applications to oxybutynin hydrochloride in pharmaceutical formulations

Uripan tablets were investigated using the proposed methods to determine the OBT in its dosage form. Acceptable percentage recoveries with standard deviations were attained according to USP [28]. These results showed no significant difference when statistically compared with those obtained by applying the BP pharmacopeia HPLC method [2] concerning Student $t$- and $F$-tests (table 4). This indicates the accuracy of the proposed methods [27].

## 3.4. Quenching mechanism

Fluorescence quenching can occur due to several molecular interactions, including static and dynamic quenching, excited-state reactions, energy transfer or molecular rearrangement [29]. Therefore, some experimental studies were conducted to identify quenching types in the proposed EY–OBT interaction. The Stern–Volmer plots were established by plotting $F^{\mathrm{o}}/F$ against $[Q]$ as described by the following equation [30]:

$$\frac{F^{\mathrm{o}}}{F} = 1 + K_{\mathrm{SV}}\ [Q],\tag{3.1}$$

where $K_{\mathrm{SV}}$ is the Stern–Volmer quenching constant and $[Q]$ is the molar concentration of the drug.

Stern–Volmer plots were linear without upward curvature ($r = 0.99$) (electronic supplementary material, figure S8), which excludes combined static and dynamic quenching where the Stern–Volmer plot is characterized by a nonlinear behaviour with an upward curvature [29]. In addition, it reveals a single type of fluorescence quenching. To determine the type of fluorescence quenching, fluorescence quenching at

**Table 4.** Application of the proposed spectrophotometric, spectrofluorometric methods and reference method for determination of OBT hydrochloride in pure form and commercial tablets.

| parameters | spectrophotometric method | | spectrofluorometric method | | official HPLC method [2] | |
|---|---|---|---|---|---|---|
| | amount taken ($\mu g\ ml^{-1}$) | found (%) | amount taken ($\mu g\ ml^{-1}$) | found (%) | amount taken ($\mu g\ ml^{-1}$) | found (%) |
| OBT hydrochloride (pure form) | 1.0 | 98.82 | 1.0 | 101.66 | 4.0 | 101.82 |
| | 2.0 | 101.97 | 3.0 | 99.08 | 10.0 | 98.40 |
| | 6.0 | 99.69 | 4.0 | 99.74 | 15.0 | 100.58 |
| | 8.0 | 99.41 | 5.0 | 100.40 | | |
| | 10.0 | 100.55 | 6.0 | 100.02 | | |
| $\bar{x} \pm$ s.d. | | 100.09 ± 1.22 | | 100.18 ± 0.96 | | 100.27 ± 1.73 |
| Student's $t$-test | | 0.26 (2.45) | | 0.09 (2.45) | | |
| variance ratio ($F$-test) | | 2.09 (6.94) | | 3.26 (6.94) | | |
| Uripan tablets (5 mg oxybutynin hydrochloride tablet) | 4.0 | 100.26 | 3.0 | 102 | 4.0 | 100.65 |
| | 6.0 | 99.69 | 4.0 | 104 | 6.0 | 99.26 |
| | 8.0 | 101.05 | 6.0 | 105.93 | 20.0 | 101.48 |
| $\bar{x} \pm$ s.d. | | 100.33 ± 0.68 | | 103.98 ± 1.96 | | 100.46 ± 1.12 |
| Student's $t$-test | | 0.17 (2.78) | | 2.69 (2.78) | | |
| variance ratio ($F$-test) | | 2.69 (19.00) | | 3.02 (19.00) | | |

Values between parentheses are the tabulated $t$- and $F$-values, respectively, at $p = 0.05$ [27].

room and elevated temperatures (295–308–323)°K were measured. By establishing Stern–Volmer plots at the raised temperatures (electronic supplementary material, figure S8), It was shown that $K_{SV}$ decreased when the temperature is elevated (electronic supplementary material, table S3), which is a typical static-quenching type, where the non-fluorescent ground state complexes formed between EY and OBT have lower stability at elevated temperatures.

Moreover, the bimolecular quenching constant ($K_q$) was calculated using the data from the Stern–Volmer plots, according to equation [29]

$$K_q = \frac{K_{SV}}{\tau_0},\tag{3.2}$$

where $K_{SV}$ is the Stern–Volmer quenching constant obtained from Stern–Volmer equations and $\tau_0$ is the fluorescence lifetime of EY = 1.1 ns in aqueous medium [31]

The calculated values of $K_q$ for OBT ranged from $1.38 \times 10^{14}$ to $0.874 \times 10^{14}\ l\ mol^{-1}\ s^{-1}$ at the temperatures being studied (electronic supplementary material, table S3).

These values confirm the static type of quenching from molecular binding and complex formation, as they are higher than $1 \times 10^{10}\ l\ mol^{-1}\ s^{-1}$, which results from the dynamic quenching.

## 3.5. Reaction thermodynamics

To assess the binding constant and number of binding sites in the proposed reaction between EY and OBT, a modified Stern–Volmer plot was established (electronic supplementary material, figure S9) by

**Scheme 1.** Suggested pathway for the reaction between OBT and EY.

plotting the relationship between $\log[(F^\circ-F)/F]$ and $\log[Q]$, as defined by the equation [32]

$$\log\left[\frac{F^\circ - F}{F}\right] = \log K_b + n\log[Q], \tag{3.3}$$

where $K_b$ is the binding constant and $n$ the number of binding sites (electronic supplementary material, table S4).

In addition, Van't Hoff graph (electronic supplementary material, figure S10) was used to study the interaction between EY and OBT by plotting $\ln K_b$ versus $1/T$ according to the equation below [33]

$$\ln K_b = -\left(\frac{\Delta H^\circ}{RT}\right) + \left(\frac{\Delta S^\circ}{R}\right), \tag{3.4}$$

where $\Delta H^\circ$ = enthalpy change, $R$ = universal gas constant = 8.314 J K$^{-1}$ mol$^{-1}$, $T$ is the absolute temperature (°K), $\Delta S^\circ$ = entropy change and $K_b$ = binding constant obtained from modified Stern–Volmer plot.

Moreover, Gibbs free energy ($\Delta G^\circ$) was calculated using the following equation [34–37]:

$$\Delta G^\circ = \Delta H^\circ - T\Delta S^\circ \tag{3.5}$$

The results are abridged in electronic supplementary material, table S4. The obtained positive $\Delta S^\circ$ value and negative $\Delta H^\circ$ value confirmed the electrostatic interaction between both ionized EY and OBT. However, positive $\Delta H^\circ$ and $\Delta S^\circ$ values indicate typical hydrophobic interactions. Negative $\Delta H^\circ$ and $\Delta S^\circ$ are characteristic of van der Waals forces and hydrogen-bond formation [34,35]. The calculated negative values of Gibb's free energy postulate the spontaneous reaction [34].

## 3.6. Determination of the reaction stoichiometry

To study the stoichiometry of the reaction between EY and OBT, Job's method [38] was used. The method was investigated using equal molar concentration for EY and OBT; the results obtained are shown in electronic supplementary material, figure S11. The results indicated a 1 : 1 ratio for EY and OBT complex composition.

In a weakly acidic medium (pH 4), electrostatic interaction occurred between the ionized hydroxyl of EY reagent and the amino group of OBT, leading to the formation of the complex [39]. The proposed pathway is outlined in scheme 1.

## 3.7. Greenness evaluation

Currently, green analytical chemistry involves the application of pharmaceutical analysis. The greenness of the proposed methods was achieved by applying 3Rs philosophy (replace, reduce and re-use) [40]. Reducing energy or heat usage and replacing organic solvents with distilled water, either by making the stock of the drug or completing the calibration's flasks, helped in obtaining a higher score in the analytical ecoscale. Analytical ecoscale is a comprehensive approach proposed to evaluate the greenness of analytical procedure; it depends on allotting penalty points to parameters of analytical processes that do not match with the ideal green analysis. The penalty points are subtracted from 100 [10]. For the proposed methods, getting an ecoscore of 95 proves 'excellent-greenness' which is better than the BP official reference method [2] that got a score of 86 (electronic supplementary material, table S5).

# 4. Conclusion

The present study develops spectroscopic methods for the quantification of OBT, in its pure form and pharmaceutical formulation. EY is regarded as an accessible, economical and safe reagent for analysis. The methods are based on the reaction between EY and the studied drug. All experimental parameters were studied and optimized using DOE, which saves time and effort. To our knowledge, all previous methods that discussed spectroscopic determination of OBT need hazardous solvents and/or heating and extraction steps. Our proposed methods offered advantages of simplicity, rapidness and green analysis. These merits make the methods convenient for quality control determining of OBT.

Data accessibility. Data are available from the Dryad Digital Repository: https://doi.org/10.5061/dryad.jwstqjq8z [41].
Authors' contributions. E.Y. performed the laboratory work, factorial model and statistical calculations. H.E., Z.A.S. and M.E.-S.M. revised the whole work and the manuscript. All authors participated in the design and study and approved the manuscript.
Competing interests. We declare we have no competing interests.
Funding. No funding supported this research.

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
