## [Peer Review File · Royal Society Open Science]

Review History

RSOS-211027.R0 (Original submission)

Review form: Reviewer 1

Is the manuscript scientifically sound in its present form?

No

Are the interpretations and conclusions justified by the results?

No

Is the language acceptable?

Yes

Do you have any ethical concerns with this paper?

No

Have you any concerns about statistical analyses in this paper?

No

Recommendation?

Reject

Comments to the Author(s)

This manuscript does not have the key element of enough novelty, and lack the necessary investigation the mechanism. Thus this manuscript does not up to standard to be accepted as an article in Royal Society Open Science.

Review form: Reviewer 2

Is the manuscript scientifically sound in its present form?

Yes

Are the interpretations and conclusions justified by the results?

Yes

Is the language acceptable?

Yes

Do you have any ethical concerns with this paper?

No

Have you any concerns about statistical analyses in this paper?

Yes

Recommendation?

Accept with minor revision (please list in comments)

Comments to the Author(s)

The manuscript entitles "Factorial design assisted spectroscopic determination of oxybutynin hydrochloride" by Eman Yosrey, et al., explained about the development of two facile spectroscopic methods for quantifying oxybutynin hydrochloride in its pure form and tablets using design of experiments. The aim of the manuscript is providing to reveal the comparison of the spectrofluorometry, spectrophotometry, voltammetry and chromatography for confirming the DOE method. The manuscript has very interesting data and is useful in utilizing of a new method for quantifying oxybutynin hydrochloride. The manuscript has been written regular with a good discussion. There are some revisions for better understanding as below:

- 1- The introduction section needs more literature survey for better literature review. Some relevant references to this research should be cited for literature survey review as follow: a) Journal of Biomolecular Structure and Dynamics, 2018, 36 (7), 1747-1763. b) The Journal of Chemical Thermodynamics, 2003, 35 (2), 199-207. c) Bulletin-Korean Chemical Society, 2001, 22 (2), 145-148. d) Journal of Molecular Structure, 2010, 979 (1-3), 227-234. e) Journal of Molecular Liquids, 2018, 256, 127-138. f) Protein and peptide letters, 2020, 27 (10), 1007-1021. g) Spectrochimica Acta Part A: Molecular and Biomolecular Spectroscopy, 2020, 228, 117528.
- 2- In Fig. 2, blank EY in acetate buffer at pH 4, in the absorbance curve, the maximum absorbance is more than 1. Can the authors cite it? A relevant reference to this subject should be cited as follow: Journal of Colloid and Interface Science, 2008, 322 (1), 119-127.
- 3- In Table 2, the intercept obtained by the spectrofluorometric method is negative. How do the authors interpret this value? Please explain in the text of manuscript.
- 4- In Table 3, in 8 microgram per milliliter, the RSD% values in spectrometric and spectrofluorometric have more different. Please interpret those different values. Two relevant references to this research should be cited for better understanding as follow: a) Journal of

Colloid and Interface Science, 2006, 299 (2), 636-646. b) Journal of Luminescence, 2018, 203, 599-608.

After the revision, the manuscript can be considered in Royal Society Open Science.

Review form: Reviewer 3

Is the manuscript scientifically sound in its present form?

No

Are the interpretations and conclusions justified by the results?

Yes

Is the language acceptable?

No

Do you have any ethical concerns with this paper?

No

Have you any concerns about statistical analyses in this paper?

No

Recommendation?

Reject

Comments to the Author(s)

From my point of view, I found the work obsolete & not novel so it is not suitable for publication in the journal.

Review form: Reviewer 4

Is the manuscript scientifically sound in its present form?

Yes

Are the interpretations and conclusions justified by the results?

Yes

Is the language acceptable?

Yes

Do you have any ethical concerns with this paper?

No

Have you any concerns about statistical analyses in this paper?

No

Recommendation?

Accept as is

Comments to the Author(s)

The manuscript is generally well written and structured, and the use of DOE in studying the affecting factors was an added value to the two proposed methods. This paper has a great potential to be accepted.

Decision letter (RSOS-211027.R0)

Dear Dr Elmansi:

Title: Factorial design assisted spectroscopic determination of oxybutynin hydrochloride
Manuscript ID: RSOS-211027

The editor assigned to your manuscript has now received comments from reviewers. We would like you to revise your paper in accordance with the referee and Subject Editor suggestions which can be found below (not including confidential reports to the Editor). Please note this decision does not guarantee eventual acceptance.

Please submit your revised paper before 08-Oct-2021. Please note that the revision deadline will expire at 00.00am on this date. If we do not hear from you within this time then it will be assumed that the paper has been withdrawn. In exceptional circumstances, extensions may be possible if agreed with the Editorial Office in advance. We do not allow multiple rounds of revision so we urge you to make every effort to fully address all of the comments at this stage. If deemed necessary by the Editors, your manuscript will be sent back to one or more of the original reviewers for assessment. If the original reviewers are not available we may invite new reviewers.

Yours sincerely,
Dr Ellis Wilde

Publishing Editor, Journals

RSC Associate Editor
Comments to the Author:
(There are no comments.)

RSC Subject Editor:
Comments to the Author:
The equations should be laid out more clearly and labelled, and figure 4 needs greater clarification in terms of both the labelling and the caption.

Reviewers' Comments to Author:

Reviewer: 1

Comments to the Author(s)

This manuscript does not have the key element of enough novelty, and lack the necessary investigation the mechanism. Thus this manuscript does not up to standard to be accepted as an article in Royal Society Open Science.

Reviewer: 2

Comments to the Author(s)

The manuscript entitles "Factorial design assisted spectroscopic determination of oxybutynin hydrochloride" by Eman Yosrey, et al., explained about the development of two facile spectroscopic methods for quantifying oxybutynin hydrochloride in its pure form and tablets using design of experiments. The aim of the manuscript is providing to reveal the comparison of the spectrofluorometry, spectrophotometry, voltammetry and chromatography for confirming the DOE method. The manuscript has very interesting data and is useful in utilizing of a new method for quantifying oxybutynin hydrochloride. The manuscript has been written regular with a good discussion. There are some revisions for better understanding as below:

1- The introduction section needs more literature survey for better literature review. Some relevant references to this research should be cited for literature survey review as follow: a)

Journal of Biomolecular Structure and Dynamics, 2018, 36 (7), 1747-1763. b) The Journal of

Chemical Thermodynamics, 2003, 35 (2), 199-207. c) Bulletin-Korean Chemical Society, 2001, 22

(2), 145-148. d) Journal of Molecular Structure, 2010, 979 (1-3), 227-234. e) Journal of Molecular

Liquids, 2018, 256, 127-138. f) Protein and peptide letters, 2020, 27 (10), 1007-1021. g)

Spectrochimica Acta Part A: Molecular and Biomolecular Spectroscopy, 2020, 228, 117528.

2- In Fig. 2, blank EY in acetate buffer at pH 4, in the absorbance curve, the maximum absorbance is more than 1. Can the authors cite it? A relevant reference to this subject should be cited as follow: Journal of Colloid and Interface Science, 2008, 322 (1), 119-127.

3- In Table 2, the intercept obtained by the spectrofluorometric method is negative. How do the authors interpret this value? Please explain in the text of manuscript.

4- In Table 3, in 8 microgram per milliliter, the RSD% values in spectrometric and spectrofluorometric have more different. Please interpret those different values. Two relevant references to this research should be cited for better understanding as follow: a) Journal of Colloid and Interface Science, 2006, 299 (2), 636-646. b) Journal of Luminescence, 2018, 203, 599-608.

After the revision, the manuscript can be considered in Royal Society Open Science.

Reviewer: 3

Comments to the Author(s)

From my point of view, I found the work obsolete & not novel so it is not suitable for publication in the journal.

Reviewer: 4

Comments to the Author(s)

The manuscript is generally well written and structured. and the use of DOE in studying the affecting factors was an added value to the two proposed methods. This paper has a great potential to be accepted.

Author's Response to Decision Letter for (RSOS-211027.R0)

See Appendix A.

RSOS-211027.R1 (Revision)

Review form: Reviewer 2

Is the manuscript scientifically sound in its present form?

Yes

Are the interpretations and conclusions justified by the results?

Yes

Is the language acceptable?

Yes

Do you have any ethical concerns with this paper?

No

Have you any concerns about statistical analyses in this paper?

No

Recommendation?

Accept as is

Comments to the Author(s)

The authors answered all of the questions as correct and the manuscript can be accepted in Royal Society Open Science.

Review form: Reviewer 3

Is the manuscript scientifically sound in its present form?

No

Are the interpretations and conclusions justified by the results?

Yes

Is the language acceptable?

No

Do you have any ethical concerns with this paper?

No

Have you any concerns about statistical analyses in this paper?

No

Recommendation?

Reject

Comments to the Author(s)

Unfortunately, the used techniques are "so obsolete" to be published in RSC. I do recommend that the authors use at least HPLC for analysis of pharmaceuticals instead of spectrophotometry that is not accurate with lack of enough sensitivity.

Review form: Reviewer 4

Is the manuscript scientifically sound in its present form?

Yes

Are the interpretations and conclusions justified by the results?

Yes

Is the language acceptable?

Yes

Do you have any ethical concerns with this paper?

Yes

Have you any concerns about statistical analyses in this paper?

No

Recommendation?

Accept as is

Comments to the Author(s)

the manuscript is well written and has a great potential to be accepted

Decision letter (RSOS-211027.R1)

Dear Dr Elmansi:

Title: Factorial design assisted spectroscopic determination of oxybutynin hydrochloride
Manuscript ID: RSOS-211027.R1

It is a pleasure to accept your manuscript in its current form for publication in Royal Society Open Science. The chemistry content of Royal Society Open Science is published in collaboration with the Royal Society of Chemistry.

Yours sincerely,
Dr Ellis Wilde
Publishing Editor, Journals

RSC Associate Editor
Comments to the Author:
According to the comments of the adjudicator, the decision was made.

RSC Subject Editor

Comments to the Author:
(There are no comments.)

Reviewer(s)' Comments to Author:

Reviewer: 2

Comments to the Author(s)

The authors answered all of the questions as correct and the manuscript can be accepted in Royal Society Open Science.

Reviewer: 3

Comments to the Author(s)

Unfortunately, the used techniques are "so obsolete" to be published in RSC. I do recommend that the authors use at least HPLC for analysis of pharmaceuticals instead of spectrophotometry that is not accurate with lack of enough sensitivity.

Reviewer: 4

Comments to the Author(s)

the manuscript is well written and has a great potential to be accepted

Appendix A

On the behalf of my co-authors, I would like to thank the editorial team members for the opportunity that we have been given to further revise our manuscript. You will find below each reviewer's comment(s) and a point to point explanation for how we dealt with the issues raised in your letter.

RSC Subject Editor

1-The equations should be laid out more clearly and labelled, and figure 4 needs greater clarification in terms of both the labelling and the caption

Reply: The equations have been laid more clearly and labelled, also figure 4 has been clarified as required by Subject Editor.

Reviewer 1:

This manuscript does not have the key element of enough novelty, and lack the necessary investigation the mechanism. Thus this manuscript does not up to standard to be accepted as an article in Royal Society Open Science

Reply: The novelty of the suggested methods originates from:

1. Applying design of experiment (DOE) which is a comprehensive approach to problem-solving that offers variance advantages over traditional one factor at a time (OFAT), these advantages include:
 - Requiring fewer trials and tests for to attain optimum results.
 - Allow studying the interaction between different factors that can't be studied in OFAT where one factor is varied and other factors are kept constant.
 - Concluding the most significant factors affecting the result, put them in order and express this information through simple and clarified tables and plots. These advantages are obtained in zero time adding time and cost saving advantages.
2. The developed methods have advantages over the reported ones in being green and simple where no need for expensive organic solvents. The only reported spectrofluorometric method for determining OBT is tedious, takes long time and lacks greenness property.

3. Applying spectrofluorometric technique provide low cost and cheap properties to the suggested method.
4. Up to our knowledge, all previous methods discussed spectroscopic determination of oxybutynin need hazardous solvents and/or heating and extraction steps. Our proposed methods offered advantages of simplicity, rapidness and green analysis.
5. The suggested procedure offered a very simple and sensitive way for determining the drug in pure and in pharmaceutical formulations. Moreover, the proposed methods utilize an inexpensive reagent, simple and cost-effective instruments which are regularly found in most quality assurance units.
6. Additionally, up till now the use of DOE wasn't yet discussed for oxybutynin determination. In this study design of experiments was used as a statistical method for assessing the influence of critical parameters on the proposed methodology to save time, effort and chemicals.

Reviewer 2:

- 1- The introduction section needs more literature survey for better literature review. Some relevant references to this research should be cited for literature survey review.

Reply: The introduction has been improved to include more literature as instructed from the reviewer. All mentioned references were included within the manuscript and considered during the revision.

- 2- In Fig. 2, blank EY in acetate buffer at pH 4, in the absorbance curve, the maximum absorbance is more than 1. Can the authors cite it? A relevant reference to this subject should be cited

Reply: Theoretically: Since $\text{absorbance} = -\log \text{transmittance}$, Absorbance value is ranged from (0-2). But practically, absorbance value is selected within (0.2-1.2) to avoid instrumental errors.

As recommended from the reviewer, the reference has been added to the maximum absorbance value.

- 3- In Table 2, the intercept obtained by the spectrofluorometric method is negative. How do the authors interpret this value? Please explain in the text of manuscript.

Reply: The negative intercept obtained by the spectrofluorometric method is expected to be interference from matrix components which may be due to OBT itself. This has been explained in details in the manuscript.

- 4- In Table 3, in 8 microgram per milliliter, the RSD% values in spectrometric and spectrofluorometric have more different. Please interpret those different values. Two relevant references to this research should be cited for better understanding

Reply: The difference in RSD% values in spectrometric and spectrofluorometric is referred to the difference in the techniques in the two methods. The absorbance values ranged from 0 to 1.0 while the fluorescence quenching values are from 0 to 1000, so RSD% obtained are quite different. All mentioned references were included within the manuscript and considered during the revision.

Reviewer 3:

From my point of view, I found the work obsolete & not novel so it is not suitable for publication in the journal.

Reply: The novelty of the suggested methods originates from:

1. Applying design of experiment (DOE) which is a comprehensive approach to problem-solving that offers variance advantages over traditional one factor at a time (OFAT), these advantages include:
 - Requiring fewer trials and tests for to attain optimum results.
 - Allow studying the interaction between different factors that can't be studied in OFAT where one factor is varied and other factors are kept constant.
 - Concluding the most significant factors affecting the result, put them in order and express this information through

simple and clarified tables and plots. These advantages are obtained in zero time adding time and cost saving advantages.

2. The developed methods have advantages over the reported ones in being green and simple where no need for expensive organic solvents. The only reported spectrofluorometric method for determining OBT is tedious, takes long time and lacks greenness property.
3. Applying spectrofluorometric technique provide low cost and cheap properties to the suggested method.
4. In this study application of experimental design methodologies to optimize the effective parameters on the quantification of oxybutynin were studied. Up to our knowledge, it is the first method for oxybutynin using quality by design. It offers advantages of being green, rapid and facile, which make it superior to previous published methods.

Reviewer: 4

Comments to the Author(s)

The manuscript is generally well written and structured. and the use of DOE in studying the affecting factors was an added value to the two proposed methods. This paper has a great potential to be accepted.

Reply: We would like to thank the reviewer for giving positive opinion in our manuscript. His revision is greatly appreciated.